# Echocardiographic parameters according to insulin dose in young patients affected by type 1 diabetes

Giacomo Zoppini[1]*, Corinna Bergamini[2], Maddalena Trombetta[1], Alessandro Mantovani[1], Giovanni Targher[1], Anna Toffalini[1], Cristina Bittante[1], Enzo Bonora[1]

**1** Section of Endocrinology, Diabetes and Metabolism, Department of Medicine, Azienda Ospedaliera Universitaria Integrata, Verona, Italy, **2** Section of Cardiology, Department of Medicine, Azienda Ospedaliera Universitaria Integrata, Verona, Italy

* giacomo.zoppini@univr.it

## Abstract

Insulin dose has been found to associate to several cardiometabolic risk factors in type 1 diabetes. Changes over time in body weight and composition may partly explain this association. However, no data are available on the relationship between insulin dose and echocardiographic parameters of both systolic and diastolic function in type 1 diabetes. Therefore, the aim of the present study was to examine systolic and diastolic echocardiographic parameters in relation to insulin dose in young patients with type 1 diabetes. The study was carried out on 93 consecutive outpatients with type 1 diabetes with a mean age of 32.8 ± 9.8 years. All patients were examined with a transthoracic echocardiography. Clinical and laboratory data were collected. The median value of daily insulin dose was used to categorized patients in two groups: high and low insulin dose group. Patients belonging to the high insulin dose group showed higher levels of cardiometabolic risk factors such as BMI, triglycerides and TG/HDL cholesterol ratio. Indexes of both systolic and diastolic function were similar in both groups except isovolumetric relaxation time (IVRT), that was significantly prolonged in patients of the high insulin group (94.4 ± 15.0 vs 86.7 ± 13.1 ms, p = 0.008). In the multivariate regression analysis, insulin dose was positively and significantly associated with IVRT. In this study we report an association between insulin dose and impaired active diastolic myocardial relaxation. Future studies are needed to further explore this observation.

## Introduction

Type 1 diabetes has been reported to increase the risk of cardiovascular morbidity and mortality [1]. Previous studies in type 2 diabetes have suggested an association between hyperinsulinemia and increased risk of cardiovascular events [2].

Recently, it has been reported an association between insulin dose and several cardiometabolic risk factors in type 1 diabetes [3]. However, despite the adverse trends of the cardiometabolic risk factors, no association has been detected between insulin dose and the incident

**Data Availability Statement:** All relevant data are within the paper.

**Funding:** The authors received no specific funding for this work.

**Competing interests:** The authors have declared that no competing interests exist.

cardiovascular events [3]. Insulin dose may change over time in relation to different factors, among them there are aging, weight gain and changes in body composition. An underlying common ground may be an increased insulin resistance that could follow body weight and composition changes [4]. Higher insulin dose has been associated with higher blood pressure, BMI, pulse rate and pulse pressure [3, 5].

No data are available on the relationship between insulin dose and echocardiographic indexes of systolic and diastolic function. Therefore, the aim of the present study was to evaluate parameters of both systolic and diastolic function according to insulin dose in a well characterized cohort of patients with type 1 diabetes.

## Materials and methods

The study was carried out on 93 consecutive outpatients with type 1 diabetes (mean age 32.8 ± 9.8 years) attending our clinic as previously described [6]. Type 1 diabetes was the inclusion criteria. The minimum age was 19 and the maximum age was 60 years. Known heart diseases was an exclusion criteria as well as pregnancy. All participants were evaluated with transthoracic echocardiography using an Esaote My Lab 70 (Esaote Spa, Genoa, Italy). The study was approved by the local ethic committee (University and Hospital Trust of Verona), and the informed consent was obtained.

Standard echocardiographic windows were employed to measure left ventricular diameters, wall thickness, end diastolic and systolic volumes of the left ventricle (LV) and left atrial volume (LAV). Mitral pulsed-wave Doppler was used to measure the A and E weave peaks. Pulse-wave tissue Doppler was performed to obtain septal and lateral mitral annulus indexes. The mean value of septal and lateral annulus E' wave was used for the analysis. Isovolumetric relaxation time (IVRT) was calculated as the time from aortic valve closure to mitral valve opening by placing the continuous Doppler between the outflow and the inflow tracts.

Clinical data, duration of diabetes, lifestyle attitudes, were collected. Body mass index was calculated by dividing weight in kilogramsa by the square of height in meters. Blood pressure was measure in supine position and hypertension was considered for values $\geq$ 140/90 mmHg. Laboratory blood measurements were determined by standard laboratory procedures on blood samples drawn in the morning after an overnight fast. Creatinine was measured using a modified Jaffé method and other biochemical blood measurements were determined by standard laboratory procedures. Hemoglobin A1c (HbA1c) was measured by high performance liquid chromatography and standardized according to IFCC. Patients with type 1 diabetes were categorized by the sex-specific median value of the daily insulin dose that was 42 U in men and 32 U in women.

## Statistical analysis

Data are summarized by means ± SD, median and proportions. Comparisons between groups were performed by t-test, non-parametric tests and $\chi$2-test for categorical variables. Bivariate linear correlations were tested by Pearson and Spearmen coefficients. Confounding factors were evaluated in a forced entry multivariate linear regression analysis. Covariates for this multivariate regression model were chosen as potential confounding factors based on their significance in univariable regression analyses or based on their biological plausibility. P-values <0.05 were considered statistically significant.

## Results and discussion

Patients with type 1 diabetes (mean age 32.8 ± 9.8 years) were categorized by the sex-specific median value of the daily insulin dose that was 42 U in men and 32 U in women. Sixteen

**Table 1. Descriptive statistics for the clinical characteristics of subjects with type 1 diabetes mellitus according to the sex-specific median insulin dose.**

|  | Under the median insulin dose (n = 43) | Above the median insulin dose (n = 50) | *p* value |
|---|---|---|---|
| Age, yrs | 34.3 ± 8.9 | 31.5 ± 10.4 | 0.172 |
| Sex, F/M | 21/22 | 20/30 | 0.259 |
| BMI, kg/m$^2$ | 22.7 ± 3.0 | 25.1 ± 3.9 | 0.001 |
| Diabetes duration, yrs | 17.2 ± 11.5 | 19.4 ± 9.0 | 0.310 |
| Systolic pressure, mmHg | 124.6 ± 12.4 | 126.8 ± 12.4 | 0.512 |
| Diastolic pressure, mmHg | 75.0 ± 7.9 | 76.6 ± 7.0 | 0.331 |
| Heart rate, bpm | 71.6 ± 11.0 | 77.0 ± 13.2 | 0.045 |
| HbA1c,% | 7.4 ± 0.9 | 8.1 ± 1.1 | 0.001 |
| Total Cholesterol, mmol/l | 4.5 ± 0.7 | 4.6 ± 0.9 | 0.467 |
| HDL cholesterol, mmol/l | 1.6 ± 0.4 | 1.4 ± 0.4 | 0.140 |
| Triglycerides, mmol/l | 0.9 ± 0.4 | 1.3 ± 0.7 | 0.002 |
| TG/HDL ratio | 1.30 ± 0.69 | 2.28 ± 1.77 | 0.001 |
| eGFRCKD-EPI, ml/min/1.73 m2 | 107.4 ± 15.0 | 110.0 ± 14.2 | 0.392 |

Values are means ± SD, percentages or percentage. BMI: body mass index. T1D: type 1 diabetes. HbA1c: glycated hemoglobin; eGFRCKD-EPI: estimated glomerular filtration rate. HDL: high-density lipoprotein. TG: triglycerides.

patients were on insulin pump and the insulin dose was lower in these patients (28.8 ± 15.4 vs 42.5 ± 19.7, p = 0.01). No association was detected between the dichotomous insulin dose variable and sex (p = 0.259). Table 1 shows that subjects with insulin dose above the median value had a higher BMI (25.1 ± 3.9 vs 22.7 ± 3.0 Kg/m$^2$, p = 0.001), a HbA1c above target as shown by the glycated hemoglobin (8.1 ± 1.1 vs 7.4 ± 0.9%, p = 0.001) and a higher level of triglycerides (1.3 ± 0.7 vs 0.9 ± 0.4 mmol/l, p = 002). The TG/HDL ratio was significantly higher in patients in the higher insulin dose group (2.28 ± 1.77 vs 1.30 ± 0.69, p = 0.001). Heart rate was slightly elevated in the group with higher daily insulin dose (77.0 ± 13.2 vs 71.6 ± 11.0 bpm, p = 0.045). The bivariate correlation analysis (Table 2) showed significant correlations of daily insulin dose with systolic and diastolic blood pressure, BMI, triglycerides levels and triglycerides to HDL-cholesterol ratio (TG/HDL ratio) only in patients of the higher insulin dose group, while a significant correlation with glycated hemoglobin was present only in patients of the low insulin dose group. IVRT did not show linear correlations with diastolic parameters, whereas it was strongly correlated with insulin dose (r = 0.29, p = 0.005).

Echocardiographic parameters in relation to daily insulin dose are reported in Table 3. Ejection fraction, LV mass, left atrial volume index (LAVI), E/A ratio and E/e' ratio did not

**Table 2. Bivariate correlations between insulin dose and cardiovascular risk factors stratified by the sex-specific median value of insulin dose.**

|  | Under the median insulin dose (n = 43) | *p* | Above the median insulin dose(n = 50) | *p* |
|---|---|---|---|---|
| Systolic blood pressure, mmHg | 0.30 | 0.055 | 0.35 | 0.012 |
| Diastolic blood pressure, mmHg | -0.02 | 0.894 | 0.40 | 0.004 |
| Heart rate, bpm | -0,10 | 0.785 | 0.25 | 0.870 |
| BMI, kg/m$^2$ | 0.16 | 0.293 | 0.29 | 0.035 |
| HDL cholesterol, mmol/l | -0.15 | 0.365 | -0.22 | 0.141 |
| Triglycerides, mmol/l | 0.05 | 0.754 | 0.30 | 0.045 |
| TG/HDL ratio | 0.05 | 0.759 | 0.33 | 0.029 |
| Glycated Hemoglobin, % | 0.39 | 0.009 | -0.13 | 0.377 |

BMI: body mass index; HDL: high density lipoprotein. TG/HDL: triglycerides to HDL cholesterol ratio.

**Table 3. Echocardiographic characteristics of type 1 diabetic patients according to the sex-specific median insulin dose.**

|  | Under the median insulin dose (n = 43) | Above the median insulin dose (n = 50) | *p* |
|---|---|---|---|
| Ventricular septal thickness, mm | 8.6 ± 1.4 | 8.8 ± 1.2 | 0.488 |
| LV mass/BSA, g/m2 | 64.9 ± 15.5 | 64.0 ± 15.0 | 0.778 |
| LA volume index, ml/m2 | 20.1 ± 4.5 | 19.4 ± 4.7 | 0.457 |
| LV ejection fraction, % | 63.4 ± 6.0 | 63.1 ± 5.0 | 0.773 |
| Septal s' velocity, cm/s | 0.110 ± 0.023 | 0.108 ± 0.021 | 0.533 |
| E/A ratio | 1.52 ± 0.39 | 1.51 ± 0.45 | 0.913 |
| E/e' ratio | 5.12 ± 1.02 | 5.17 ± 1.19 | 0.833 |
| IVRT, ms | 86.7 ± 13.1 | 94.4 ± 15.0 | 0.008 |

Values are means ± SD or percentages. LV: left ventricular. ˚LV mass/BSA was calculated using the following formula: (0.8*(1.04*(LVDd+IVSd+PWd)3-(LVDd)3))
+0.6/body surface area. IVRT: isovolumetric relaxation time.

differ between the two groups. IVRT was significantly higher in patients in the higher daily insulin dose group (94.4 ± 15.0 vs 86.7 ± 13.1 ms, p = 0.008). To further analyze the factors associated with the IVRT, we performed a multivariate linear regression model with IVRT as the dependent variable (Table 4). Insulin dose was positively and significantly associated with IVRT (p = 0.005). The results remained the same when insulin dose to weight (U/Kg/day) was used.

In this study on type 1 diabetes outpatients, different findings emerged in relation to the daily insulin dose. Consistently with the sex related insulin sensitivity [7], men showed a higher median value of the daily insulin dose with respect to women. In the higher insulin dose group, BMI, heart rate, glycated hemoglobin, TG/HDL ratio and triglycerides were significantly elevated compared to the low insulin dose group. A significant correlation was found between daily insulin dose and blood pressure, BMI, triglycerides and TG/HDL ratio only in the group with higher insulin dose. While, glycated hemoglobin was significantly correlated to daily insulin dose only in patients in the lower insulin dose group. However, the new and more interesting findings of our study were related to the daily insulin dose and echocardiographic parameters. IVRT, a marker of active diastolic ventricular relaxation [8], was significantly higher in the group with higher insulin dose, while septal thickness; LV mass/BSA; LAVI; LV ejection fraction; septal s' velocity; E/A ratio; E/e' ratio; did not differ between the two groups.

The association between insulin dose and cardiometabolic risk factors is consistent with a previous study, that showed similar findings in type 1 diabetes [3]. Of interest is the correlation

**Table 4. Multivariate linear regression analysis with IVRT as the dependent variable in patients with type 1 diabetes.**

|  | Standardized beta coefficient | *p* |
|---|---|---|
| Insulin dose, U/day | 0.374 | 0.005 |
| Age, years | 0.042 | 0.749 |
| BMI, kg/m$^2$ | -0.075 | 0.542 |
| Diabetes duration, years | -0.141 | 0.282 |
| Glycated Hemoglobin, % | 0.047 | 0.709 |
| Heart rate (bpm) | -0.259 | 0.024 |
| TG/HDL cholesterol ratio | 0.205 | 0.104 |

BMI: body mass index; HDL: high density lipoprotein; bpm: beat per minute.

we observed between insulin dose and glycated hemoglobin that was significant only in patients in the low insulin dose group. Patients in the higher insulin dose have a higher body weight and this means that they are more insulin resistant than patients in the low insulin dose group [9]. These correlations seem to suggest that after a certain threshold of insulin resistance, the relation between insulin dose and glycated hemoglobin reaches a sort of plateau. This relation between insulin dose and glycated hemoglobin is intriguing and may be clinically relevant. It could imply that in our patients with type 1 diabetes we should pursue different approaches, such as a stronger control of the body weight or the use of hypoglycemic agents with different mechanisms of action, in order to ameliorate glucose control instead of increasing insulin dose as suggested in a recent study [10].

At the best of our knowledge this is the first observation of a prolonged IVRT in patients with a higher daily insulin dose compared to patients with a low insulin dose. Interestingly, the association between insulin dose and IVRT persisted in the multivariate model after confounders adjustments. Moreover, IVRT prolongation is the only alteration we observed among other indexes of both systolic and diastolic function and this may be due to the fact that IVRT is an index of LV active diastolic relaxation that precedes LV filling [11].

IVRT occurs when both aortic and mitral valves are closed, thus it is characterized by changes in myocardial elasticity, while ventricular volume remains constant [11, 12]. Impaired heart relaxation involves increased ventricular wall stiffness and abnormal cardiac filling [12]. The explanation of the observed association between insulin dose and IVRT prolongation is not straightforward, nevertheless different mechanisms, not mutually exclusive, may be explored. First of all it should be remarked that patients with a higher insulin dose showed a significantly higher body weight, triglycerides, pulse rate and glycated hemoglobin. These alterations may indicate a higher insulin resistance level, a higher activation of sympathetic activity, higher levels of circulating insulin and higher mean blood glucose levels. All the above alterations may impact on the myocardium structure and function [13].

A first mechanism could be an interference of the calcium handling. Calcium handling disturbances appear to be a key feature of diabetic cardiomyocytes [14], consisting in a prolonged SERCA2 mediated Ca2+ removal from cytosol during diastole [14]. A reduced activity and expression of SERCA2 protein have been described in heart failure and diabetic cardiomyopathy [15]. Altered calcium handling may directly alter cardiac relaxation [16]. However, the precise role of SERCA2 downregulation in the pathophysiology of diabetic cardiomyopathy is not yet well described. Insulin resistance and obesity may affect calcium handling and thus alter IVRT [17, 18]. Moreover, increased sympathetic activity may be the cause of the impairment of IVRT as so nicely shown after renal denervation [19]. Concerning our results, it is interesting to remember that hyperinsulinemia, that may occur in patients with a high insulin dose, may cause sympathetic activation [20]. Furthermore, the results of a recent study that showed improved diastolic function, preserved calcium handling and attenuated myocardial insulin resistance after empagliflozin treatment, are of extreme interest [21]. These results were obtained in a hyperinsulinemic animal model of type 2 diabetes. Treatment with empagliflozin significantly reduced the levels of plasma insulin [21].

A second mechanism could be related to cardiac hypertrophy, one of the first structural changes in the diabetic cardiomyopathy [22]. Insulin resistance, hyperactivity of the sympathetic system and hyperinsulinemia were all linked to cardiac hypertrophy [23]. Altered IVRT may be a precocious alteration in the development of cardiac hypertrophy [24]. Brzyżkiewicz et al, in their study showed a positive correlation between insulin dose and BNP levels [25]. Insulin, when elevated, may promote cardiac hypertrophy [24]. This may be a direct effect of insulin through its own receptor or by binding to the igf-1 receptor, which presents large homology with the insulin receptor [26, 27]. Animal models have shown that chronic

hyperinsulinemia may induce cardiac hypertrophy and fibrosis [28, 29]. In a pressure overload mice model, depletion of insulin, obtained with streptozotocin treatment, attenuates systolic dysfunction [29]. Of note, treatment with insulin improved hyperglycemia during pressure overload on one side, while increased cross-sectional area and decreased relative vascular density on the other, thereby exacerbating cardiac hypoxia leading to cardiomyocytes death and inducing heart failure [29]. Moreover, a positive relationship between cardiac hypertrophy and plasma insulin concentration has been reported [30]. An interesting result of our study is that patients in the higher insulin dose group showed significantly elevated level of triglycerides [31]. It has been reported that in the setting of hyperglycemia, insulin resistance and hypertriglyceridemia, the myocardium ability to use glucose as an energy source is impaired and a switch to use free fatty acids occurs [32]. This switch may compromise the anti-oxidant capacity of the myocardium thus leading to cardiac dysfunction [32].

And finally a third mechanism could concern the content of glycogen of the myocardium. An excess in cardiac glycogen accumulation has been associated to hypertrophy and dysfunction [33]. The occurrence of cardiac glycogen accumulation in human diabetic patients and animal diabetic models is known from many years [34–36]. Recently, it has been shown that a sustained exposure to insulin and high glucose concentration can induce glycogen deposition through a process defined "cardiomyocytes glycophagy" [37]. This phenomenon may be relevant to our study. In fact, it has been shown that a long and sustained exposure to insulin only in a high glucose media leads to glycogen accumulation in cardiomyocytes [37]. This experimental model can recall the situation of the group of patients with higher insulin dose that also showed higher glycated hemoglobin. Thus, the situation of this group can suggest a long exposure to insulin in a high glucose milieu.

Interestingly, a recent study in patients with hereditary haemochromatosis, a cardiomyopathy with iron accumulation, has shown a significant reduction of IVRT after a year course of venesection [38].

The present study has limitations, it is a cross-sectional study thus does not allow to infer on the cause-effect relationship. It is a single center study. Due to the low number of patients on insulin pump we could not differentiate the effect of basal-bolus treatment from insulin pump treatment. We cannot exclude a bias effect of the unmeasured variables. Our study has however various strengths, the number of patients was quite high, moreover they were a homogenous group in fairly good metabolic control and with a very few and not advanced chronic complications.

## Conclusions

In this study we report a correlation between insulin dose and impaired active diastolic myocardial relaxation. Future studies are needed to further investigate the mechanisms underlying this observation.

## Author Contributions

**Conceptualization:** Giacomo Zoppini, Corinna Bergamini.

**Data curation:** Giacomo Zoppini, Corinna Bergamini, Anna Toffalini, Cristina Bittante.

**Formal analysis:** Giacomo Zoppini.

**Methodology:** Corinna Bergamini.

**Writing – original draft:** Giacomo Zoppini.

**Writing – review & editing:** Maddalena Trombetta, Alessandro Mantovani, Giovanni Targher, Enzo Bonora.

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
