## [Decision Letter · Decision Letter 0]

26 Oct 2020

PONE-D-20-20811

Echocardiographic parameters according to insulin dose in young patients affected by type 1 diabetes.

PLOS ONE

Dear Dr. Zoppini,

Thank you for submitting your manuscript to PLOS ONE. After careful consideration, we feel that it has merit but does not fully meet PLOS ONE’s publication criteria as it currently stands. Therefore, we invite you to submit a revised version of the manuscript that addresses the points raised during the review process.

We look forward to receiving your revised manuscript.

Kind regards,

Yoshihiro Fukumoto

Academic Editor

PLOS ONE

Journal Requirements:

2.Thank you for including your ethics statement:  "The study was approved by the local ethic committee, and the informed consent was obtained".   

3. You indicated that you had ethical approval for your study. In your Methods section, please ensure you have also stated whether you obtained consent from parents or guardians of the minors included in the study

4. We note you have included a table to which you do not refer in the text of your manuscript. Please ensure that you refer to Table 4 in your text; if accepted, production will need this reference to link the reader to the Table.

Reviewers' comments:

Reviewer's Responses to Questions

**Comments to the Author**

1. Is the manuscript technically sound, and do the data support the conclusions?

Reviewer #1: Yes

Reviewer #2: Yes

2. Has the statistical analysis been performed appropriately and rigorously? 

Reviewer #1: Yes

Reviewer #2: Yes

3. Have the authors made all data underlying the findings in their manuscript fully available?

Reviewer #1: Yes

Reviewer #2: Yes

4. Is the manuscript presented in an intelligible fashion and written in standard English?

Reviewer #1: Yes

Reviewer #2: Yes

5. Review Comments to the Author

Reviewer #1: This is an interesting study to evaluate the association between diastolic myocardial relaxation (IVRT) and insulin-dose. The authors observed positive and significant association between insulin dose and IVRT even after adjusting some clinical factors. I have the several comments to the authors.

1. Why the authors focus on IVRT? Did the authors see any associations between other diastolic parameters and insulin-dose? Some diastolic parameters may be significant after adjusting for confounders.

2. Table 4: In the multivariable model, how did the authors select the variables? AIC or BIC based? Stepwise selection? High-dose group has higher heart rate that affects IVRT. I have a concern about the authors detected the difference of heart rate (HR) between high-dose and low-dose groups. The association between IVRT and insulin-dose significant further adjusted for HR?

Reviewer #2: Thank you for allowing me to review this paper titled “Echocardiographic parameters according to insulin dose in young patients affected by type 1 diabetes”.

This is a cross-sectional study to assess the association between insulin dose and active diastolic myocardial relaxation.

I recommend the manuscript for publications and have a few comments for revision to share:

The aim is to assess the association between cardiac function insulin dose in “young patients” with type 1 diabetes, but there is no mention of age in the abstract.

In the method section, in addition to age, the median insulin dose (u/kg) needs to be provided.

Introduction. I think it will be helpful as well as important to categorize the evidence on cardiovascular events in type 1 vs. since the data is controversial.

Method: Despite citing reference 4, some background about the setting, date range, and patient population will be helpful.

I suggest being more specific about: “Clinical and laboratory data were collected”.

Results:

Instead of “worse glycemic control,” I suggest using HbA1C “above target”.

Was there any patient on a closed-loop system?

Was there any difference in insulin use based on the type of treatment (MDI vs. pump) even with the current sample size?

Can you please provide some info on why the authors decided to use the median to categorize the insulin usage?

Table 3. Does using the variables included in table 4 in the multivariate regression models lead to bias for “unmeasured” variables? Would it be possible to include Table 1 variables in the model?

Discussion: strength: the authors have described the limitations of the study since this analysis indicates a correlation and not a cause-effect relationship.

6. PLOS authors have the option to publish the peer review history of their article (what does this mean?). If published, this will include your full peer review and any attached files.

Reviewer #1: No

Reviewer #2: No

---

## [Author Response · Author response to Decision Letter 0]

29 Oct 2020

PONE-D-20-20811

Echocardiographic parameters according to insulin dose in young patients affected by type 1 diabetes.

PLOS ONE

A) The financial disclosure was not changed. No figures.

 A) Done following the style requirements.

2.Thank you for including your ethics statement: "The study was approved by the local ethic committee, and the informed consent was obtained". 

A) done

 A) done

3. You indicated that you had ethical approval for your study. In your Methods section, please ensure you have also stated whether you obtained consent from parents or guardians of the minors included in the study

 A) no minors were included in the study.

4. We note you have included a table to which you do not refer in the text of your manuscript. Please ensure that you refer to Table 4 in your text; if accepted, production will need this reference to link the reader to the Table.

A) Thanks. We report the refer of table 4 in the text.

Reviewers' comments:

Reviewer's Responses to Questions

Comments to the Author

1. Is the manuscript technically sound, and do the data support the conclusions?

Reviewer #1: Yes

Reviewer #2: Yes

2. Has the statistical analysis been performed appropriately and rigorously?

Reviewer #1: Yes

Reviewer #2: Yes

3. Have the authors made all data underlying the findings in their manuscript fully available?

Reviewer #1: Yes

Reviewer #2: Yes

4. Is the manuscript presented in an intelligible fashion and written in standard English?

Reviewer #1: Yes

Reviewer #2: Yes

5. Review Comments to the Author

Reviewer #1: This is an interesting study to evaluate the association between diastolic myocardial relaxation (IVRT) and insulin-dose. The authors observed positive and significant association between insulin dose and IVRT even after adjusting some clinical factors. I have the several comments to the authors.

Thanks for your comments.

1. Why the authors focus on IVRT? Did the authors see any associations between other diastolic parameters and insulin-dose? Some diastolic parameters may be significant after adjusting for confounders.

A) No linear correlations were detected between IVRT and diastolic parameters. IVRT was significantly correlated to insulin dose. We add a sentence in the results section. Insulin dose was not an independent predictor of E/A, E/e’ or LAVI.

2. Table 4: In the multivariable model, how did the authors select the variables? AIC or BIC based? Stepwise selection? High-dose group has higher heart rate that affects IVRT. I have a concern about the authors detected the difference of heart rate (HR) between high-dose and low-dose groups. The association between IVRT and insulin-dose significant further adjusted for HR?

A) In the statistical section we specified how the variables were chosen. The model was a forced entry model. Thanks for your important suggestion about heart rate. We included heart rate in the model and the IVRT maintained its significance. We report this new model, that includes heart rate, in the revised manuscript and the results were changed accordingly.

Reviewer #2: Thank you for allowing me to review this paper titled “Echocardiographic parameters according to insulin dose in young patients affected by type 1 diabetes”.

Thanks for your comments.

This is a cross-sectional study to assess the association between insulin dose and active diastolic myocardial relaxation.

I recommend the manuscript for publications and have a few comments for revision to share:

The aim is to assess the association between cardiac function insulin dose in “young patients” with type 1 diabetes, but there is no mention of age in the abstract.

In the method section, in addition to age, the median insulin dose (u/kg) needs to be provided.

A) We added mean age of patients in the abstract section and mean age plus median insulin dose in the methods section. 

Introduction. I think it will be helpful as well as important to categorize the evidence on cardiovascular events in type 1 vs. since the data is controversial.

A) We changed the paragraph of the introduction and added two more references. We hope that the controversy is more evident.

Method: Despite citing reference 4, some background about the setting, date range, and patient population will be helpful.

I suggest being more specific about: “Clinical and laboratory data were collected”.

A) We added more information in the method section.

Results:

Instead of “worse glycemic control,” I suggest using HbA1C “above target”.

A) we change the expression.

Was there any patient on a closed-loop system?

A) 16 subjects were on pump.

Was there any difference in insulin use based on the type of treatment (MDI vs. pump) even with the current sample size?

A) thanks for this question. We compared the insulin dose between patients on pump and MDI, patients on pump had a lower insulin dose. We add the results in the result section.

Can you please provide some info on why the authors decided to use the median to categorize the insulin usage?

A) We use the median value because there is not a recognized cutoff level of insulin dose. In the multivariable model we use insulin dose as a continuous variable.

Table 3. Does using the variables included in table 4 in the multivariate regression models lead to bias for “unmeasured” variables? Would it be possible to include Table 1 variables in the model?

A) We add a sentence in the limitation of the study to acknowledge the possible effect of unmeasured variables. 

Discussion: strength: the authors have described the limitations of the study since this analysis indicates a correlation and not a cause-effect relationship.

A) thanks for the comment.

6. PLOS authors have the option to publish the peer review history of their article (what does this mean?). If published, this will include your full peer review and any attached files.

Do you want your identity to be public for this peer review? For information about this choice, including consent withdrawal, please see our Privacy Policy.

Reviewer #1: No

Reviewer #2: No

---

## [Decision Letter · Decision Letter 1]

11 Dec 2020

Echocardiographic parameters according to insulin dose in young patients affected by type 1 diabetes.

PONE-D-20-20811R1

Dear Dr. Zoppini,

We’re pleased to inform you that your manuscript has been judged scientifically suitable for publication and will be formally accepted for publication once it meets all outstanding technical requirements.

Kind regards,

Yoshihiro Fukumoto

Academic Editor

PLOS ONE

Additional Editor Comments (optional):

Reviewers' comments:

Reviewer's Responses to Questions

**Comments to the Author**

1. If the authors have adequately addressed your comments raised in a previous round of review and you feel that this manuscript is now acceptable for publication, you may indicate that here to bypass the “Comments to the Author” section, enter your conflict of interest statement in the “Confidential to Editor” section, and submit your "Accept" recommendation.

Reviewer #1: All comments have been addressed

Reviewer #2: All comments have been addressed

2. Is the manuscript technically sound, and do the data support the conclusions?

Reviewer #1: Yes

Reviewer #2: Yes

3. Has the statistical analysis been performed appropriately and rigorously? 

Reviewer #1: Yes

Reviewer #2: Yes

4. Have the authors made all data underlying the findings in their manuscript fully available?

Reviewer #1: Yes

Reviewer #2: Yes

5. Is the manuscript presented in an intelligible fashion and written in standard English?

Reviewer #1: Yes

Reviewer #2: Yes

6. Review Comments to the Author

Reviewer #1: (No Response)

Reviewer #2: The authors have responded to the comments. I have no additional comments to the revised version of the manuscript.

7. PLOS authors have the option to publish the peer review history of their article (what does this mean?). If published, this will include your full peer review and any attached files.

Reviewer #1: No

Reviewer #2: No

---

## [Editor Report · Acceptance letter]

15 Dec 2020

PONE-D-20-20811R1 

Echocardiographic parameters according to insulin dose in young patients affected by type 1 diabetes. 

Dear Dr. Zoppini:

I'm pleased to inform you that your manuscript has been deemed suitable for publication in PLOS ONE. Congratulations! Your manuscript is now with our production department. 

Kind regards, 

on behalf of

Dr. Yoshihiro Fukumoto 

Academic Editor

PLOS ONE